# Distinct transcriptomes and autocrine cytokines underpin maturation and survival of antibody-secreting cells in systemic lupus erythematosus

Weirong Chen[1,5], So-Hee Hong[1,2,5], Scott A. Jenks [1], Fabliha A. Anam[1], Christopher M. Tipton[1], Matthew C. Woodruff[1], Jennifer R. Hom[1], Kevin S. Cashman[1], Caterina Elisa Faliti[1], Xiaoqian Wang[1], Shuya Kyu[3], Chungwen Wei [1], Christopher D. Scharer [4], Tian Mi [4], Sakeenah Hicks[4], Louise Hartson[1], Doan C. Nguyen [3], Arezou Khosroshahi[1], Saeyun Lee[1], Youliang Wang[1], Regina Bugrovsky[1], Yusho Ishii[1], F. Eun-Hyung Lee [3] ✉ & Ignacio Sanz [1] ✉

Systemic lupus erythematosus (SLE) is an autoimmune disease characterized by multiple autoantibody types, some of which are produced by long-lived plasma cells (LLPC). Active SLE generates increased circulating antibody-secreting cells (ASC). Here, we examine the phenotypic, molecular, structural, and functional features of ASC in SLE. Relative to post-vaccination ASC in healthy controls, circulating blood ASC from patients with active SLE are enriched with newly generated mature CD19⁻CD138⁺ ASC, similar to bone marrow LLPC. ASC from patients with SLE displayed morphological features of premature maturation and a transcriptome epigenetically initiated in SLE B cells. ASC from patients with SLE exhibited elevated protein levels of CXCR4, CXCR3 and CD138, along with molecular programs that promote survival. Furthermore, they demonstrate autocrine production of APRIL and IL-10, which contributed to their prolonged in vitro survival. Our work provides insight into the mechanisms of generation, expansion, maturation and survival of SLE ASC.

Plasma cells (PCs) represent a critical effector immune cell type responsible for both protective and pathogenic antibody responses. PC exhibit significant diversity in terms of homeostasis, phenotype, location and longevity[1]. Indeed, the PCs compartment can be envisioned as a heterogeneous collection of antibody-secreting cell populations (ASC) that share essential properties, including constitutive antibody secretion, morphology, expression of licensing transcription factors (TF), such as Blimp-1, XBP1, and IRF4, silencing of PAX5, and in humans, high levels of CD38 and CD27 expression with or without CD19[1,2]. ASC sharing these identifiers can be differentiated into Ki-67⁺ plasmablasts (PB), which appear in the circulation approximately seven days after acute antigenic stimulation and undergo apoptosis within 2 weeks (short-lived PB), and resting PCs, which can persist for years in the BM in mice even without stimulating antigen[3].

[1]Department of Medicine, Division of Rheumatology, Lowance Center for Human Immunology, School of Medicine, Emory University, Atlanta, GA, USA. [2]Department of Microbiology, Ewha Womans University, Seoul, Republic of Korea. [3]Department of Medicine, Division of Pulmonary, Allergy, Critical Care and Sleep Medicine, School of Medicine, Emory University, Atlanta, GA, USA. [4]Department of Microbiology and Immunology, School of Medicine, Emory University, Atlanta, GA, USA. [5]These authors contributed equally: Weirong Chen, So-hee Hong. ✉e-mail: f.e.lee@emory.edu; ignacio.sanz@emory.edu

Our previous study demonstrated that human long-lived plasma cells (LLPC), responsible for maintaining antibodies against measles and mumps for decades after infection, reside within a CD19⁻ CD138⁺ PC population in the human BM[1].

Systemic lupus erythematosus (SLE) is a systemic autoimmune disease in which defective B-cell tolerance leads to the production of large amounts of pathogenic autoantibodies, some of which persist throughout the disease with stable serum titers even after sustained depletion of precursor B cells. These long-lived autoantibodies include anti-RNA-binding proteins (RBP), such as Smith/RNP, and anti-Ro. In contrast, other SLE autoantibodies, including anti-dsDNA, anti-ribosomal P, and 9G4 antibodies, fluctuate with disease activity[4] and immunosuppressive therapy[5,6]. The behavior of persistent serum autoantibodies can be explained by the accumulation of B cell-independent BM LLPC. In addition, local PC are a major contributor to the pathogenesis and outcome of lupus nephritis, a critical disease manifestation. Despite their central pathogenic role and a growing understanding in animal models, the biology of PC in human SLE is still poorly understood and understudied. Therefore, our ability to therapeutically modulate PC in this disease is quite limited. This is a critical unmet need, as B cell depletion can only impede the new generation of PC while failing to eliminate pre-formed pathogenic LLPC.

In this study, to address these important knowledge gaps, we present a systematic analysis of peripheral blood ASC in patients with SLE, and compare their phenotypic and morphological characteristics, molecular regulatory programs, and survival properties with those of healthy adults following immunization. Our results reveal that diverse ASC populations in patients with SLE share common precursors, undergo enhanced peripheral maturation and display coordinated expression of CXCR3 and CXCR4. Notably, a major component of the SLE ASC transcriptome originate from gene sets epigenetically imprinted B cells from patients with SLE[7]. Moreover, SLE ASC are endowed with prolonged survival, mediated at least partly, by the transcriptional regulation of intrinsic apoptotic programs. Furthermore, SLE ASC display elevated levels of CD138 and autocrine production of APRIL and IL-10.

## Results

### Complexity and magnitude of circulating ASC in active SLE
Flow cytometry was used to characterize the abundance and diversity of circulating ASC in SLE, relative to the newly generated ASC in the circulation of healthy subjects after recall immunizations. As depicted in Fig. 1a, SLE patients exhibited a greatly increased abundance of total ASC relative to vaccinated healthy controls (vax-HC) and patients with inactive SLE. These results align with previous studies, including our own research[4,8]. Notably, this study employed simultaneous evaluation of multiple phenotypic markers, yielding insights into the heterogeneity of expanded ASC, especially the more mature CD19⁻ compartment, which had remained understudied in SLE. Building upon our previous investigation of PC in healthy control (HC)[1,2], we identified four distinct populations of IgD⁻ CD27⁺⁺ CD38⁺⁺ ASC based on the expression of CD19 and CD138: Pop 2 (CD19⁺ CD138⁻), Pop 3 (CD19⁺ CD138⁺), Pop 4 (CD19⁻ CD138⁻), and Pop 5 (CD19⁻ CD138⁺) (Fig. 1b). Importantly, the levels of CD138 expression were significantly higher in CD138⁺ ASC populations in SLE relative to their counterparts in Vax-HC (Fig. 1b, c). We had described an earlier pre-PB population (Pop 1: CD19⁺ IgD⁻ CD27⁻ CD38⁺)[2,9], which is distinguished from transitional and pre-germinal center (GC) cells by the absence of CD10 and CD24, markers that were not included in the current analysis[9], and Pop 1 was thus not quantified here. Overall, both total ASC and CD19⁺ and CD19⁻ ASC fractions were increased in terms of relative frequencies and absolute numbers in both SLE groups relative to steady-state HC. Active SLE also displayed higher numbers of both ASC fractions relative to vax-HC and inactive SLE. However, in inactive SLE, only CD19⁻ ASC showed a significant elevation relative to vax-HC (Fig. 1d). Among

the four ASC populations, the most pronounced expansions were observed for CD19⁻ Pop 4/5 between active SLE and vax-HC, although significant differences were also detected for CD19⁺ Pop2/3 between these two groups (Figs. 1e, 2). Furthermore, significant differences were present between active and inactive SLE, except for Pop 4 (Fig. 1e), an ASC population that, as illustrated by the expression of various markers, appears to be heterogeneous in composition and may include fractions outside the ASC lineage. Overall, the contribution of CD19⁻ Pop 4/5 to the circulating ASC pool was greatly increased in both SLE groups relative to vax-HC, where they are scarcely detected, consistent with previous studies[2,10] (Fig. 2). Noteworthy, the abundance of both circulating CD19⁺ and CD19⁻ ASC fractions, as well as individual ASC populations, correlated significantly with SLE disease activity, with the strongest correlations observed for circulating CD19⁺ ASC and Pops 2, 3 and 5. (Supplementary Fig. 1).

### Immune phenotype of circulating ASC in active SLE
To better understand the nature of circulating SLE ASC, we examined the expression of signature regulators of PC differentiation and survival including Blimp-1, BCMA and IL-6R (Fig. 3a, Supplementary Fig. 2). As in our previous studies, a distinctive level of expression of Blimp-1, a TF driving the differentiation of B cells into PC[11,12], was detected at consistently high levels in over 90% of all ASC populations except for Pop 4, a notable fraction of which lacked Blimp-1 expression. BCMA and IL-6R, which play crucial roles in ASC survival[13,14], exhibited a similar pattern of expression, further confirming the heterogeneity of Pop 4.

ASC can be categorized as proliferative, immature PB, or quiescent, mature PC, with the former population enriched in active peripheral immune responses, and the latter predominantly found in the stable, pre-formed BM PC compartment[1]. The majority of SLE ASC demonstrated ongoing or recent proliferation as indicated by Ki-67 expression at levels comparable to PB in vax-HC[15]. Of note, a predominance of Ki-67⁺ cells were present in all SLE ASC populations, both in active and inactive disease, although inactive SLE exhibited a significantly larger fraction of Ki-67⁻ cells in all ASC populations (Fig. 3b). Consistent with other studies[4], Ki-67 was tightly associated with HLA-DR expression, a marker that is downregulated during PC maturation. Interestingly, uncoupling of Ki-67 and HLA-DR was observed in a substantial fraction of inactive SLE ASC, suggesting a more heterogeneous compartment in these patients (Fig. 3b, Supplementary Fig. 3a). Overall, active SLE and vax-HC ASC, the latter serving as a control for newly generated ASC, displayed a homogenous Ki-67⁺/HLA-DR⁺ phenotype, even in cells with an otherwise mature BM-like phenotype, i.e., CD19⁻CD138⁺ fraction (Pop 5).

The ability of ASC to migrate to non-lymphoid tissues or the BM is determined by the expression of CXCR3 and CXCR4, chemokine receptors that promotes migration to inflammatory tissues induced by Th1 responses as well and bone marrow and spleen respectively[16–19]. CXCR4 was expressed in the majority of circulating SLE ASC at significantly higher levels than vax-HC and inactive SLE (Fig. 3c, Supplementary Fig. 3b). In contrast, CXCR3 was overexpressed in active SLE compared to inactive SLE, with a similar but not statistically significant trend observed relative to vax-HC. Notably, the majority of ASC in active SLE co-expressed CXCR3 and CXCR4, whereas dual expression of these receptors was present at very low levels (<10%) in both inactive SLE and vax-HC. Supporting the role of CXCR4 in ASC homing and retention in the BM, even in the pro-inflammatory milieu of the lupus BM[20], this receptor was expressed in the majority of BM PC whereas CXCR3 expression was rare on SLE BM PC (Fig. 3d).

Previous studies have reported that in steady-state HC, the majority of circulating ASC are IgA-producing PB, with a surge of antigen-specific IgG ASC observed 6–7 days after systemic immunization[21,22]. In the context of SLE, it has been reported that IgA ASC represent the majority of expended (up to 90%; average 58%), in

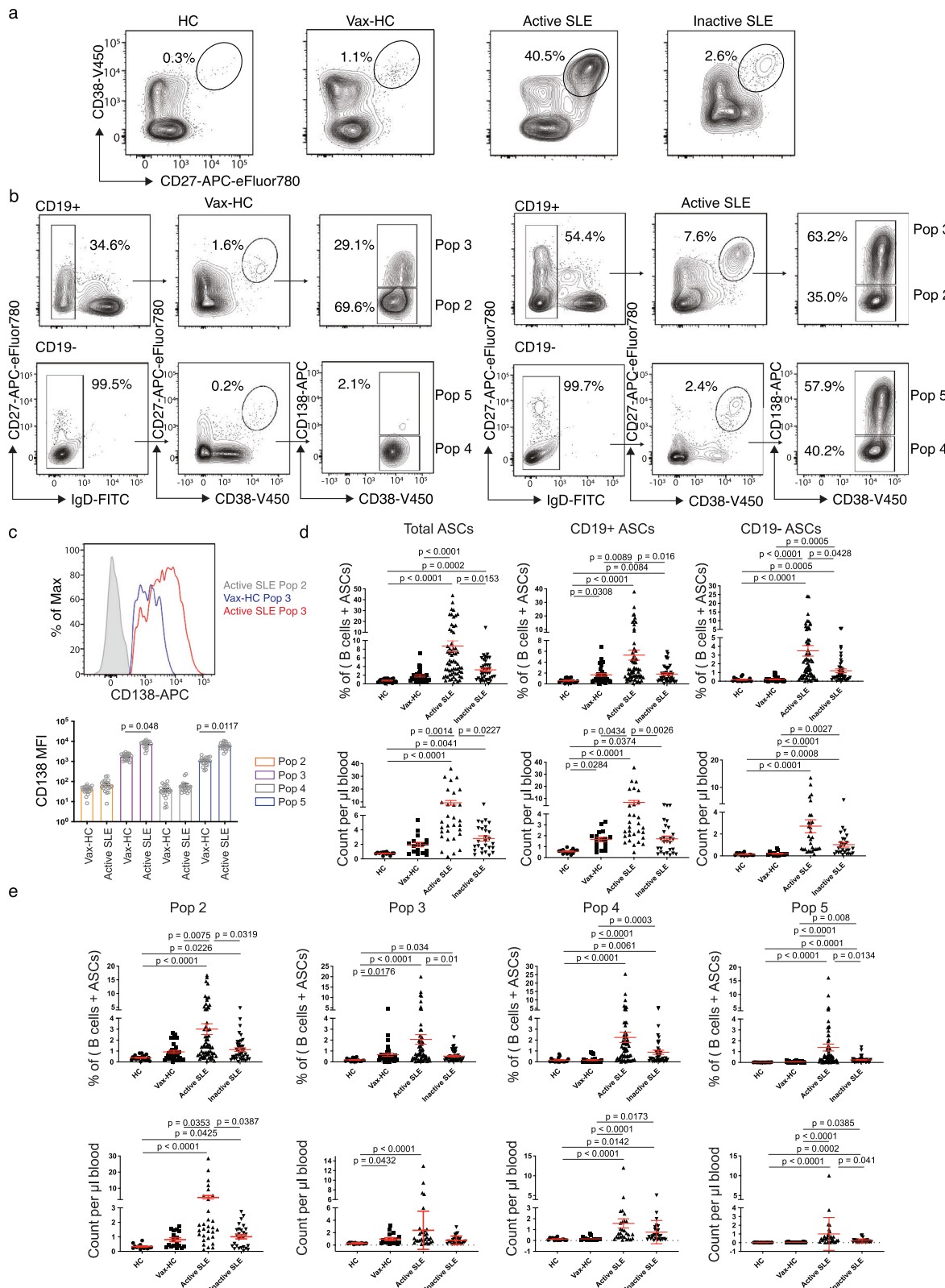

European patients with low disease activity[23]. In our predominantly African American cohort, patients with active SLE exhibited a dominance of circulating IgG ASC over IgA ASC, with the highest ratio observed within Pop 5. Conversely, IgG/IgA ASC ratios were consistently lower in inactive SLE patients. Furthermore, IgG/IgM ASC ratios were significantly increased in active SLE compared to inactive

SLE (Supplementary Fig. 4a, b). Notably, in inactive SLE, IgA ASC were increased in the Ki-67⁻ fraction relative to Ki-67⁺ fraction (Supplementary Fig. 4c). Overall, ASC expansions in active SLE were primarily composed of IgG-producing cells, while IgA ASC predominated in inactive SLE, replicating the characteristic profile of acute immune responses and steady-state conditions, respectively.

**Fig. 1 | Complexity and magnitude of circulating antibody-secreting cells in active SLE.** Peripheral blood mononuclear cells (PBMCs) were isolated from steady-state healthy donors, influenza vaccinated heathy subjects on day 7 post immunization, active SLE patients, or inactive SLE patients. Isolated cells were stained and analyzed by flow cytometry. **a** Representatives plots of the frequencies of CD19[+] ASC (CD27[++] CD38[++]) in the CD3[-] CD14[-] CD19[+] gate. **b** Gating strategy for the identification of ASC populations Pops 2-5, based on the CD19[+] and CD19[-] gate, in active SLE patients (left) and heathy subjects on day 7 post influenza immunization (right). **c** Expression of surface CD138 on ASC populations from active SLE patients ($n = 21$) and influenza vaccinated heathy subjects ($n = 20$). Histograms (top), illustrate representative examples and box plots include compiled data (bottom). Data are shown as mean ± SEM. Statistical significance between SLE patients and vaccinated healthy subjects was assessed using Student's *t* test. **d** Frequencies of total ASC, CD19[+] ASC, and CD19[-] ASC in total B cells and ASC

combined (top) from steady-state healthy donors ($n = 19$), influenza vaccinated heathy subjects ($n = 33$), active SLE patients ($n = 63$), or inactive SLE patients ($n = 41$). Numbers of total ASC, CD19[+] ASC, and CD19[-] ASC per µL of the peripheral blood (bottom) from steady-state healthy donors ($n = 14$), influenza vaccinated heathy subjects ($n = 17$), active SLE patients ($n = 30$), or inactive SLE patients ($n = 27$). Data are shown as mean ± SEM. **e,** Frequencies of each ASC Pops 2−5, in the total B cells and ASC combined (top) from steady-state healthy donors ($n = 19$), influenza vaccinated heathy subjects ($n = 33$), active SLE patients ($n = 63$), or inactive SLE patients ($n = 41$). Numbers per µL of the peripheral blood (bottom) from steady-state healthy donors ($n = 14$), influenza vaccinated heathy subjects ($n = 17$), active SLE patients ($n = 30$), or inactive SLE patients ($n = 27$). Data are shown as mean ± SEM. Statistical significance was assessed using Kruskal−Wallis test followed by Dunn's test for multiple pairwise comparisons (**d**, **e**). *p* values are shown on plots. Source data are provided as a Source Data file.

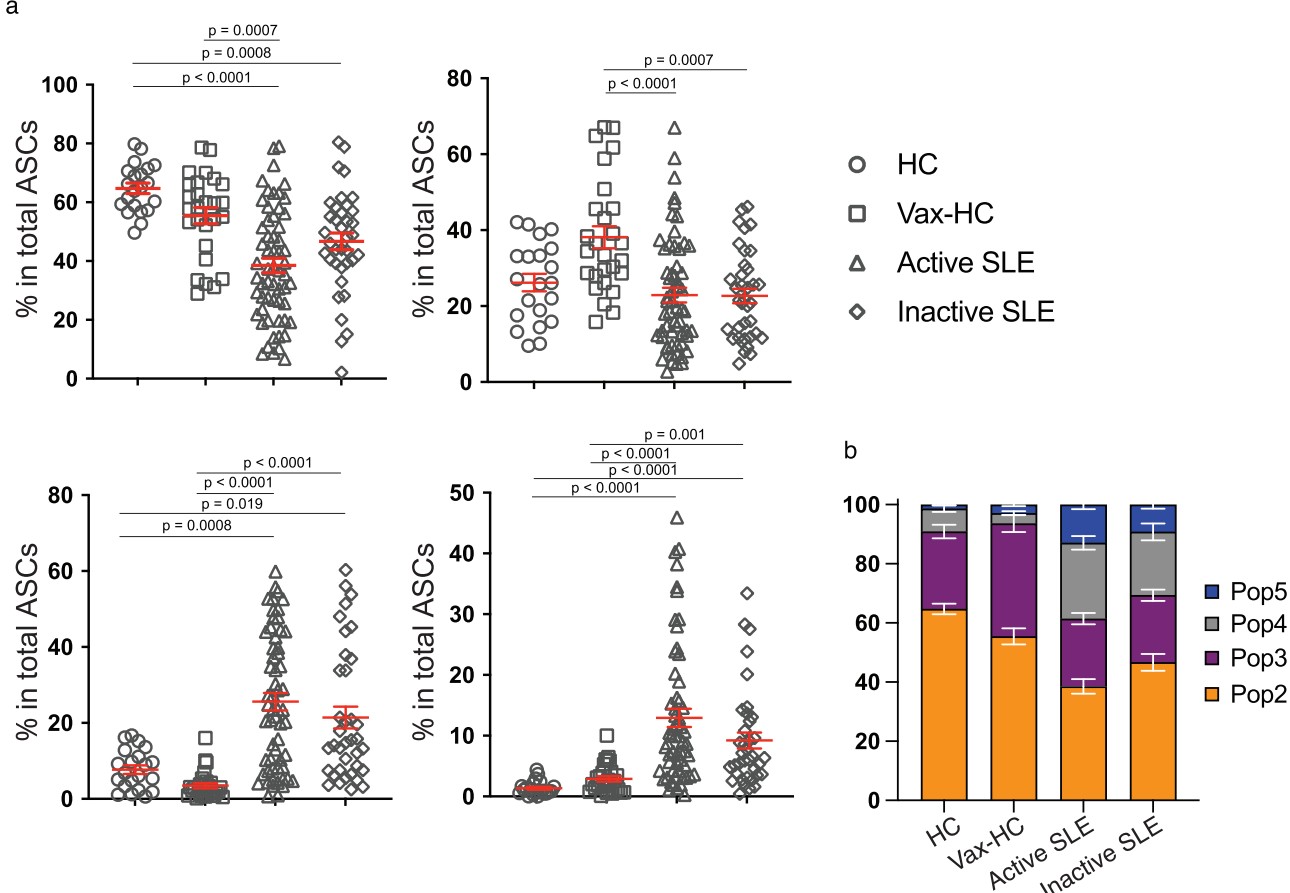

**Fig. 2 | Increased CD19[-] ASC substantially contribute to the circulating ASC pool in active SLE. a** Composition of each ASC population in total circulating ASC in steady-state healthy donors ($n = 21$); influenza vaccinated heathy subjects on day 7 post immunization ($n = 27$); active SLE patients ($n = 59$); and inactive SLE patients ($n = 37$). Data are shown as mean ± SEM. Statistical significance was assessed using

Kruskal−Wallis test followed by Dunn's test for multiple pairwise comparisons. *p* values are shown on plots. **b** Stacked bars showing the distribution of each circulating ASC population, Pop 2, Pop 3, Pop 4 and Pop 5, for the indicated groups. Data are shown as mean ± SEM in grouped analysis. Source data are provided as a Source Data file.

## Peripheral ASC display enhanced mature morphology in patients with SLE

One remarkable finding from our studies was the significant expansion of CD19[-] ASC, particularly the CD19[-]CD138[+] fraction (Pop 5), which displayed a surface phenotype resembling more mature BM PC and is typically scarce in normal vaccination responses[1]. Nevertheless, the large majority of ASC in active SLE, including Pop 5, demonstrated features of recent generation and proliferation, as indicated by the coordinated expression of HLA-DR and Ki-67. Morphologically, these ASC in active SLE exhibited characteristic maturational features

(Fig. 4a–c). Thus, similar to mature PC in the healthy BM, all SLE ASC populations showed a higher cytoplasm/nucleus ratio (Fig. 4b)and prominently displayed an increased number of cytoplasmic vacuoles (Fig. 4c), which are typical of mature BM PC[1]. Electron Microscopy (EM) studies further confirmed the presence of vacuolar structures together with an expanded endoplasmic reticulum (ER) structure in ASC from active SLE, particularly in Pop 3 from SLE compared to vax-HC (Fig. 4d, e). Notably, the level of ER complexity of peripheral blood Pop 5 from SLE patients was equivalent to that of the most mature HC BM pop D, which represents a major reservoir of human long-lived PC[24].

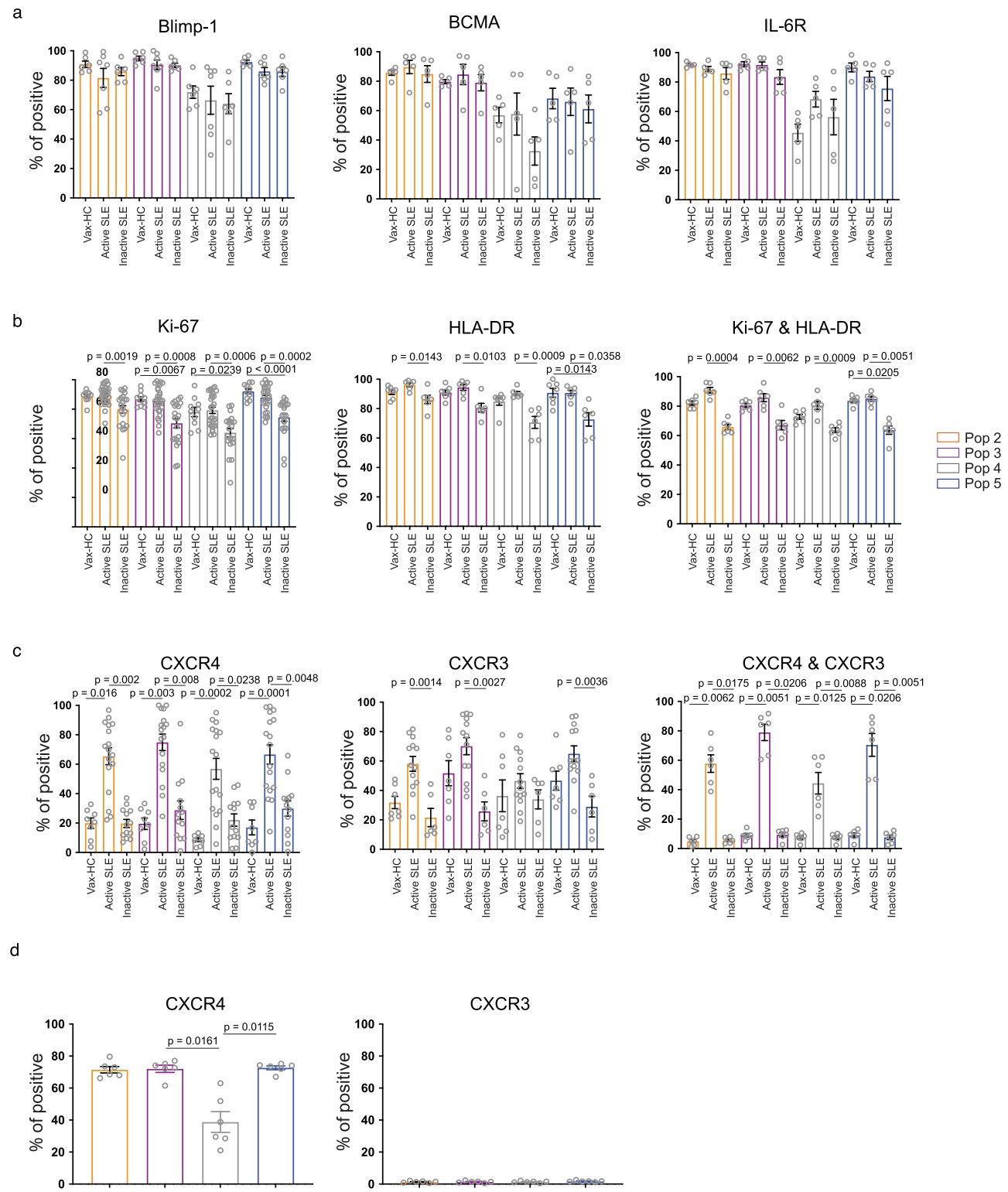

## Heterogeneous ASC responses in patients with SLE share common precursors

Whether different types of ASC, including immature PB and mature PC, arise from distinct B cell precursors remains to be fully understood. Also unknown is whether a common precursor might generate different ASC progeny through separate differentiation pathways, or instead, through sequential maturation. To address this question, we performed next generation sequencing of the antibody repertoire of ASC populations in active SLE (Supplementary Table 1). The results of sequencing of Variable, Diverse and Joining regions (VDJ) confirmed our previous findings, showing ASC in most SLE patients exhibited a predominantly polyclonal repertoire[8]. However, in some individual patients, the ASC repertoire was dominated by large clonal expansions that were detectable in all populations, providing an informative window into a common cellular origin (Fig. 5a and Supplementary Fig. 6). Shared clones were observed among ASC populations (Fig. 5b, c) and quantified through the Morisita index (Fig. 5d) in both types of patients. A large proportion of ASC clones detected in the

**Fig. 3 | Immune phenotype of circulating ASC in active SLE. a–c** Peripheral blood mononuclear cells (PBMCs) were isolated from steady-state healthy donors, influenza vaccinated heathy subjects on day 7 post immunization, active SLE patients, or inactive SLE patients and isolated cells were stained and analyzed by flow cytometry. **a** Expression of intracellular molecular Blimp-1 in ASC populations from vaccinated heathy subjects ($n = 6$), active SLE patients ($n = 7$), or inactive SLE patients ($n = 6$); and surface molecules BCMA ($n = 5$ per group) and IL-6R ($n = 5$ per group). Data are shown as mean ± SEM. **b** Expression of intracellular Ki-67 in ASC populations from vaccinated heathy subjects ($n = 10$), active SLE patients ($n = 28$), or inactive SLE patients ($n = 21$), surface HLA-DR on ASC populations from vaccinated heathy subjects ($n = 7$), active SLE patients ($n = 6$), or inactive SLE patients ($n = 6$) and their co-expression on ASC populations ($n = 6$). Data are shown as mean ± SEM. **c** Expression of surface expression of CXCR3 from vaccinated heathy subjects ($n = 7$), active SLE patients ($n = 13$), or inactive SLE patients ($n = 6$); CXCR4 from vaccinated heathy subjects ($n = 8$), active SLE patients ($n = 17$), or inactive SLE patients ($n = 13$), and their co-expression on ASC populations ($n = 6$). Data are shown as mean ± SEM. **d** Bone Marrow mononuclear cells (BMMCs), were isolated from SLE patients, and analyzed by flow cytometry for the expression of CXCR4 (left) and CXCR3 (right) on BM PC populations, Pop A-D, from SLE patients ($n = 6$). Data are shown as mean ± SEM. Statistical significance was assessed using Kruskal-Wallis test followed by Dunn's test for multiple comparisons within the same ASC population (**a-d**). $p$ values are shown on plots. Source data are provided as a Source Data file.

CD19[−] Pops 4/5 were present in the CD19[+] Pops 2/3, and likewise, substantial connectivity were observed between ASC populations with or without CD138 expression (Fig. 5d). Notably, ASC clones identified as abnormally expanded, defined by a difference threshold of 0.1% larger than the previous clone in size-ranked clones[8], displayed higher degrees of inter-population connectivity (Supplementary Fig. 7a). Even within the same-day blood samples, which inherently underestimate connectivity between cellular compartments due to unsynchronized development and/or different clonal persistence[8], the more mature CD19[−] CD138[+] (Pop 5) displayed up to 14% global connectivity with the more immature CD19[+] CD138[−] (Pop 2) among expanded clones. Among the 25 largest clones present in Pop 5, 35% to 40% of clones were also documented in Pop 2 (Fig. 5e). Overall, the load of somatic hypermutation (SHM) was similar across all populations with mean frequencies of 6–8% for isotype switched ASC and a significant fraction of all Pops displaying SHM rates below 5% (Supplementary Fig. 7b). These results are consistent with our previous studies demonstrating a major contribution of newly activated extrafollicular naive B cells as well as preexisting memory cells to the expansion of ASC during SLE flares[8]. The contribution of extrafollicular naive B cells is also consistent with the degree of SHM we previously documented in naive-derived ASC within 10 days of acute SARS-CoV2 infection[25,26]. Importantly, despite the high clonal connectivity between ASC populations, no identifiable intraclonal sequence divergence or sequential progression of somatic mutation was observed. These findings are consistent with extrafollicular ASC differentiation whether from newly recruited naive B cells or long-standing memory cells.

## ASC from patients with SLE express a distinct transcriptome

We had previously reported the transcriptional and epigenetic programs of peripheral and BM ASC in HC[2,27–29]. We had also demonstrated that SLE B cells are characterized by an epigenetically determined transcriptional program dominated by a set of TF[7,30,31]. Here, we performed RNA-seq of sorted ASC populations to gain insights into disease-specific developmental and survival programs that could contribute to the observed phenotypic and morphological differences between SLE and HC ASC. A substantial number of differentially expressed genes (DEGs) were identified between the corresponding populations in active SLE relative to vax-HC (Fig. 6). In general, SLE and HC ASC were primarily separated by a first principal component, which contributed 21% of the total DEGs identified (Fig. 6a). Consistent with our previous findings in HC[2,29], the physiological ASC transcriptome suggested a transition from highly similar Pops 2/3 to Pop 5, with a gradual extinction of certain transcripts and acquisition of a larger set of DEGs, resulting in a significant number of DEGs between HC Pop 2 and 5 (Fig. 6b, d). In contrast, consistent with the enhanced maturation indicated by phenotypic and morphological features, the most significant transcriptional difference in SLE ASC was observed between Pop 2 and Pop 3, which in turn showed high similarity to Pop 5. This pattern suggests that peripheral SLE ASC undergo accelerated maturation that appears to be already completed within the CD19+ compartment (Fig. 6b–d). The largest difference between SLE and vax-

HC was detected in Pop 3, with the majority of DEGs representing overexpressed genes in SLE (Fig. 6c, d). More than 100 DEGs were found to be shared across all SLE ASC populations relative to vax-HC (Fig. 6e), including interferon (IFN)-dependent genes and a collection of highly immunologically relevant IFN-independent genes (Fig. 6f). Notably, these overlapping DEGs included a subset of 69 genes that are initially dysregulated in SLE naive B cells including a set of epigenetically imprinted TF, largely responsible for the SLE-associated B cell transcriptome (Fig. 6e, f)[7,30,31].

Gene set enrichment analysis (GSEA) identified signaling pathways that exhibited consistent hyperactivity across all SLE ASC populations, predominantly type I and Type II IFN. This generic SLE ASC pattern also indicated increased activity in TNF-α signaling, inflammatory responses, estrogen responses, and IL-6 signaling. These SLE-associated pathways were particularly prominent in Pops 3/5, which were also significantly enriched for IL-2/STAT5 signaling, and anti-apoptotic programs, as further discussed below. On the other hand, the generic vax-HC ASC profile was characterized by enhanced glycolysis, fatty acid metabolic programs, UV response and TGF-β signaling (Supplementary Fig. 8a). In SLE, TNF-α and other inflammatory pathways were predominantly pronounced in SLE Pops 2/3 relative to vax-HC, and more attenuated, albeit still overexpressed in the more mature SLE Pop 5 (Supplementary Fig. 8b). KEGG analysis corroborated the findings from GSEA pathways, and revealed the overexpression of additional innate immune pathways, including TLR[32,33], NOD[34,35] and RIG-I[36], all of which have significance in SLE and other autoimmune diseases, and we had previously observed to be upregulated in SLE B cells[7,8,30] (Supplementary Fig. 8c).

Notably, in SLE ASC Pops 3 and 5, there was a significant increase in the expression of genes associated with ASC differentiation and/or survival. These included IL-10 (previously only associated with ASC differentiation)[37,38], APRIL (TNFSF13)[13,39,40], as well as anti-apoptotic genes, adhesion and homing markers (CD31/PECAM, CD54/ICAM1, CD69, and CXCR4), and BCR signaling components including NR4A1 (Nur77)[41] and Lyn[42–44], JUNB and FOS, and ATF3 (Fig. 6g). Critically, SLE ASC expressed a distinctly pro-survival program with strong upregulation of anti-apoptotic genes and coordinated downregulation of downregulation of pro-apoptotic genes (Fig. 7a). Overexpressed anti-apoptotic genes prominently included Bcl-2 and Bcl-2-family member Mcl-1, both central to inhibit PC death. Correspondence between increased transcription and elevated protein levels was documented for BCL-2 by intracellular flow cytometry (Fig. 7b). In turn, concurrently downregulated pro-apoptotic genes included caspases, BAD, BAX, BIK and FADD[45]. Notably, BCL-2 expression was most pronounced in SLE Pop 2 and remained at high levels in SLE Pops 3/5. In contrast, high expression of MCL1 was acquired by SLE Pops 3/5. Additionally, these two populations showed high expression of TNF receptor superfamily member 10c and 10d, which serve as decoy receptors and inhibit TRAIL-induced apoptosis[46] (Fig. 7a). Finally, SLE ASC also shared with SLE B cells upregulation of the AP-1 TF Jun and Fos which we have recently reported to be greatly upregulated in bone marrow long-lived

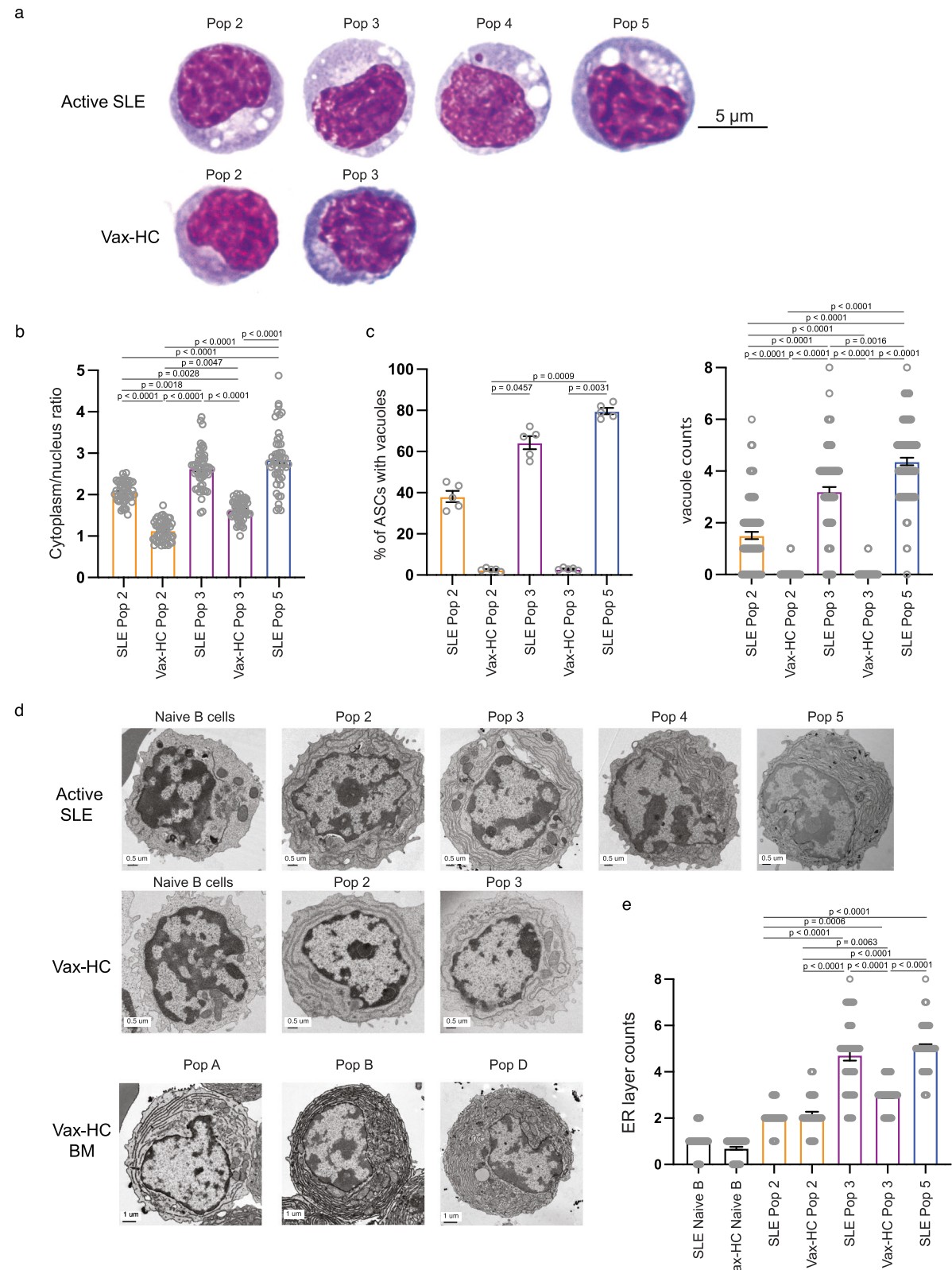

PCs[47]. Moreover, Jun is also known to play an essential role in the survival and drug resistance of Multipe Myeloma PCs[48].

### ASC from patients with SLE display enhanced survival in vitro

To validate the transcriptomic disparities in survival programs, we sorted ASC from the blood of active SLE patients and vax-HC to assess their survival ability in our in vitro model using mesenchymal stomal cells secretome[49]. In addition to promoting ASC survival, these conditions optimize the antibody production function of purified ASC, which is typically compromised by the sorting process, thereby resulting in initially increased Elispot numbers in the absence of proliferation[15,27,49]. Compared to vax-HC ASC, SLE ASC displayed higher viability with significantly slower decay even at day 3 of culture, a finding consistent with resistance of SLE ASC to the characteristic

**Fig. 4 | Peripheral SLE ASC display enhanced mature morphology. a–c** Wright-Giemsa staining of FACS-sorted peripheral ASC populations from active SLE patients ($n = 5$) and influenza vaccinated heathy subjects on day 7 post immunization ($n = 5$). **a** Representative images of Wright-Giemsa staining of the 100X magnification of ASC. The ratio of cytoplasm to nucleus ($n = 50$) (**b**), the percentage of ASC populations with vacuoles in each subject ($n = 5$), and the vacuoles in each cell ($n = 100$) was evaluated (**c**). Data are shown as mean ± SEM. Statistical significance was assessed by Kruskal-Wallis test with multiple comparisons using Dunn's test among all groups (**b**, **c**). **d–e** Electron microscopy of ASC populations sorted from influenza vaccinated heathy subjects and active SLE patients. Representative images (**d**) and compiled data for density of ER layers ($n = 50$) (**e**) in circulating ASC. Images of mature bone marrow PCs from vaccinated heathy subjects were also included for reference (**d**). Data are shown as mean ± SEM. Statistical significance was assessed using Kruskal–Wallis test with multiple comparisons performed by Dunn's test among all groups excluding Naive B-cell groups (**e**). $p$ values are shown on plots. Source data are provided as a Source Data file.

pro-apoptotic properties of normal ASC. Substantially increased survival of SLE ASC was extended over 28 days of culture with 50% viability, in contrast to almost complete disappearance of Vax-ASC by day 14 (Fig. 7c). Subsequent culture studies provided further evidence for the enhanced secretion of IL-10 and APRIL by SLE ASC, in line with the corresponding transcriptional levels (Fig. 7d). Crucially, the survival of SLE ASC was significantly compromised when cultured in the presence of blocking antibodies against IL-10, APRIL, or both. In contrast, cultures treated with antibody isotype controls exhibited no notable impact on ASC viability. These findings underscore a major role for these autocrine loops in sustaining the survival of SLE ASC (Fig. 7e).

## Discussion

Given the central role of autoantibodies in SLE, it is critical to understand their mechanistic basis. The overall abundance of peripheral ASC (PB) correlates with disease activity as evidenced by phenotypic and modular transcriptional analyses[50,51]. It has also been postulated that autoreactive ASC (PC) may be increased in the SLE BM, since high titers of certain autoantibodies persist even after B cell depletion, suggesting sustained production by autonomous LLPC. These features could be explained by increased PC generation during active disease, accompanied by enhanced maturation and homing to the BM, as well as prolonged survival mediated by intrinsic programming and/or extrinsic cues. However, these questions have not been fully addressed due to the limitations of existing studies, which have employed unfractionated samples containing various ASC (including PB and PC), utilized limited surface markers, and lacked comprehensive RNA sequencing[52].

In order to address these knowledge gaps, we present the first in-depth characterization of the multiple circulating ASC populations present in the blood of SLE patients, comparing them to ASC that typically surge 7 days after recall immunization in HC but subsequently regress to normal levels within 2 weeks, a profile that has been ascribed to programmed apoptosis that regulates the duration of the acute immune response. It has also been proposed that circulating ASC may represent, at least in part, the displacement of preexisting BM PC by new arrivals competing for survival niches during recall responses[53].

Our findings confirm the major increase in the global amount of circulating ASC in active SLE, while also provide the first evidence of a substantial expansion of CD138+ cells lacking CD19 expression, a phenotype typically ascribed to mature BM LLPC[1,10]. The increased generation of CD19− ASC has important implications for the SLE treatment strategies involving anti-CD19 agents, including monoclonal antibodies[54] and CD19 CAR-T cells[55,56]. However, in contrast to terminally differentiated resting BM PC, even the most differentiated circulating SLE ASC display a Ki-67+/HLA-DR+ phenotype akin to immature ASC observed in early immunization responses[53]. In keeping with newly published evidence in the mouse[57], this profile argues against BM displacement as a major contributor to the expansion of peripheral ASC in active SLE. Notably, in inactive SLE, a significant fraction of ASC lacked expression of Ki-67 and/or HLA-DR, with an average of 40% of all ASC populations lacking both markers. This finding could be explained either by equal displacement of all BM PC populations, or by the chronic generation and prolonged peripheral persistence of newly produced ASC in inactive SLE. The latter scenario could also account for the prevalence of IgA ASC in inactive disease,

similar to steady-state HC[22,23]. This contrasts with the IgG dominance observed in active SLE and in HC following systemic immunization, thereby suggesting distinct sources of ASC in different SLE states, with a predominance of low-grade proliferation of housekeeping mucosal IgA in inactive disease. Further investigation is needed to elucidate the extended phenotype of homing markers, antibody repertoire, and antigen-specificity to confirm this possibility.

Regardless, VDJ repertoire sequencing clearly demonstrates a large degree of clonal sharing across all ASC populations in active SLE. Together with the lack of intraclonal accumulation of SHM, these findings demonstrate the existence of shared B cell precursors with the potential to generate ASC at different stages of maturation. While the definitive evidence for this or the alternative model of early branching of common precursors into separate ASC populations would require in-depth longitudinal single cell analysis of antigen-specific lymph node reactions, when combined with the progressive transcriptional changes observed, and the known ability of CD19+ ASC to differentiate into CD19− ASC[58] and our new single cell analysis of bone marrow PC maturation[59], our data favor a model of longitudinal maturation acting upon a common precursor.

SLE ASC display several phenotypic features that provide important insights into their abnormal functions, which might underlie the multiple pathogenic aspects associated with the disease. In addition to their enhanced generation and higher numbers of mature CD19− ASC, multiple properties contribute to their accumulation and survival in the BM. This process eventually leads to the large reservoir of autoantibody-producing PC, which explains the presence of stable autoantibodies characteristic of SLE, prominently including anti-RNA-binding antibodies such as anti-Ro and anti-Smith/RNP. Notably, and in contrast to previous studies of SLE ASC[52], and of Sjogren's syndrome ASC[60], SLE ASC exhibit distinctive characteristics, including high levels of the chemokine receptors CXCR4 and CXCR3, largely in combination. CXCR4 plays a central role in the homing of PC to the BM and their retention in long-lived niches. It may also contribute to the migration and retention of ASC in Lupus kidneys[17,61], an activity also promoted by CXCR3 induced by Th1-like responses[19]. Therefore, the enhanced co-expression of CXCR4 and CXCR3 in circulating SLE ASC suggests a strong capability of these cells to migrate into inflamed Lupus kidneys, whether as a primary or a secondary amplifying event. Of note, increased CXCR4 transcription was closely paralleled by the transcriptional levels of the RBP ZFP36L, with maximal expression observed in SLE ASC Pop 3 and 5. ZFP36L deficiency is known to limit the abundance of molecules involved in ASC homing to the BM, and its absence leads to diminished accumulation of ASC in the BM[62].

Other mechanisms contributing to increased generation and survival of SLE ASC included their enhanced expression of CD138 and the elevated production of IL-10 and APRIL. IL-10 is a potent mediator of B cell proliferation and PC differentiation and can be produced by various cell types in SLE, including peripheral Th10 CD4 T cells[37,38,63]. Our data demonstrate that SLE ASC can also produce IL-10, creating an autocrine loop. IL-10 production has been associated with regulatory functions in PC[64], therefore it appears counterintuitive that it may be increased in SLE ASC. However, previous studies have also demonstrated increased B cell activity with IL-10[65], potential therapeutic benefits of IL-10 inhibition in SLE[66,67]; pro-inflammatory effects of IL-10 in an IFN-dominated milieu[68]; and role of IL-10 as autocrine factor in

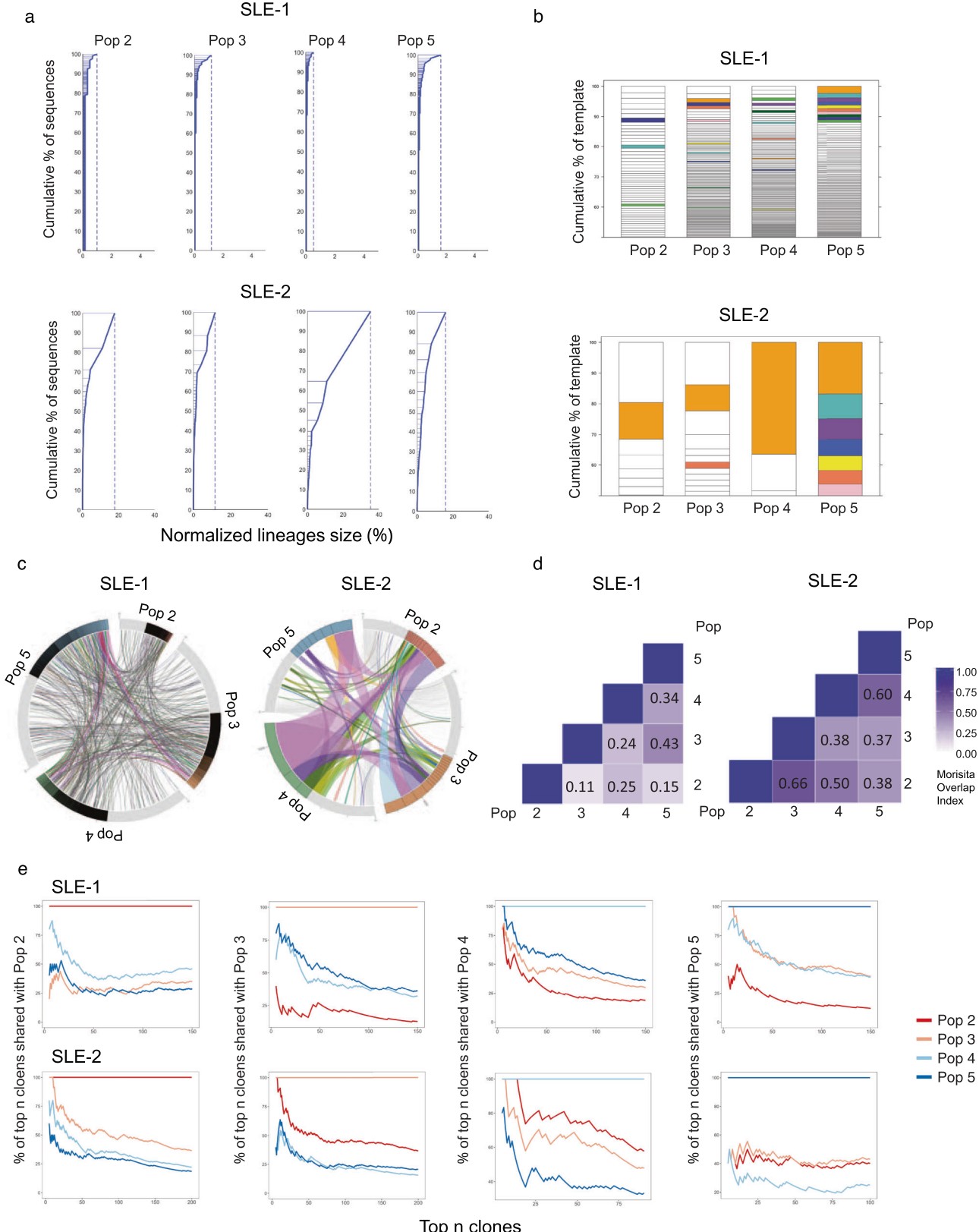

supporting B cell differentiation into ASC[69], suggesting a complex role for IL-10 in autoimmune diseases. Furthermore, our findings provide the first evidence of autocrine production of APRIL by ASC, which is substantially increased in SLE and enhances ASC survival in vitro. These results are consistent with previous publication about the autocrine role of APRIL on B cell differentiation[70]. Acting through TACI

and BCMA receptors, APRIL is an important regulator of PC differentiation and survival[39,71], as also demonstrated in our previous work with a BM mimetic culture system[49]. Notably, CD138 was overexpressed in SLE ASC, and CD138-bound APRIL induces ASC differentiation independently of IFN, a mechanism that is enhanced in SLE[40]. Therefore, our results support the existence of an autocrine effect of

**Fig. 5 | Heterogeneous SLE ASC responses share common precursors.** AIRR-seq was used to analyze the VDJ heavy chain repertoire of ASC populations from 2 active SLE patients. **a** Clonality of the repertoire in ASC populations is shown by plotting normalized lineage size versus the cumulative percent of sequences. Lineages are size-ranked in descending order along the extent of the y-axis representing 100% of all the sequences. Horizontal lines delineate the individual lineages. **b** Stacked bar plots demonstrate the diversity of the repertoire by showing descending, size-ranked clones as segments comprising percentages of the total repertoire. The largest 10 clones of the reference population pop 5 are colored, and like-colors in other populations show identical clones in other populations. **c** Circos plot shows interconnectedness of the ASC populations by plotting the sequences from each population in clonal size-ranked order with the largest clones being the most clockwise portion of each population segment. Lines between ASC populations indicate matched clones between ASC populations. **d** The Morisita Overlap Index demonstrates the similarity of repertoires in various ASC populations as a value from 0 (no similarity) to 1 (identical repertoire). The color strength is indicative of interconnectivity. **e** The clonal relatedness of SLE ASC populations is shown by plotting the percent of shared clones with Pop 2, Pop 3, Pop 4, and Pop 5, respectively (y axis), within the top numbers of clones (x axis).

APRIL production by SLE ASC, which may be further enhanced by the higher levels of CD138 expression observed in SLE. This autocrine loop could be of significant relevance to SLE pathogenesis and treatment, as highlighted by the therapeutic benefits of agents targeting components, such as TACI-Ig and anti-BCMA strategies, including BCMA CAR-T-cell therapy[72].

Our study provides original insights into the transcriptional regulation of SLE ASC, demonstrating a major transcriptional inflection point in SLE ASC Pop 3. Overall, a distinct SLE transcriptome emerges, characterized by an inflammatory signature driven by type I and type II IFN, IL-6, and TNF, and mediated by NF-κB signaling. A fraction of this signature was also shared by naive B cells in SLE. These findings, combined with our previous SLE epigenetic studies, support the idea of an epigenetically determined transcriptional program propagated throughout B-cell differentiation. Of particular interest are the shared overexpression of TRL7 and EPSTI1 (epithelial mesenchymal interacting protein 1) in SLE naive B and ASC. TLR7 hyperactivity plays a major role in SLE development, including as monogenic B cell-intrinsic gain-of-function somatic mutations[73], and in the activation of naive and DN2 B cells, which leads to the generation of pathogenic ASC through extrafollicular pathway[30]. Additionally, we had shown that EPSTI1 is highly demethylated in SLE naive B cells, with a strong strong correlation between demethylation and disease activity[7]. Interestingly, EPSTI1 is also overexpressed in Sjogren's B cells and induces B cell activation and antibody production through NF-κB signaling[74]. SLE ASC, especially Pops 3/5, exhibit overexpression of AP-1 proteins JunB and ATF3 as well as EGR genes. We had previously identified ATF3 and EGR as the top TF overexpressed in SLE B cells owing to enhanced chromatin accessibility[7]. Combined, ATF3 facilitates the hetero-trimerization of AP-1 proteins, and JunB is essential for cell identity in other immune cells[75]. JunB also plays a central role in multiple myeloma cell proliferation, and drug resistance in the BM microenvironment[48], and serves as a key regulator of myeloma BM angiogenesis[76]. Hence, these JunB-mediated mechanisms could contribute to the fate determination, accumulation, and resistance to treatment of SLE ASC. Finally, SLE ASC, particularly Pop 2, exhibit a strikingly high level of TOX2 expression. TOX2 is a TF that induces T follicular helper cells differentiation through enhanced BCL6 chromatin accessibility[77,78]. TOX2 is also important for inducing and sustaining GC formation and promoting T-bet expression in B cells, which is most prominent in activated SLE naive B cells and DN2 cells, both representing ASC precursors[8,30].

In this study, we also found that SLE ASC overexpress additional innate immune pathways, including TLR, NOD, and RIG-I, which we had previously characterized as hyperactive in SLE DN2 cells, one of the main precursors of ASC in SLE[7,8,30]. Furthermore, TLR, NOD, TNF pathways, other pro-inflammatory cytokines, and BCR signaling could contribute to ASC stimulation through the canonical NF-κB signaling pathway[79]. Combined, our work provides original evidence for the transcriptional programming of SLE ASC through adaptive and adaptive immune pathways other than the type I IFN demonstrated in earlier work[52]. On the other hand, the vax-HC ASC exhibit enriched glycolysis and fatty acid metabolic pathways. This observation may be consistent with a larger fraction of recently generated PB[80–82]. However, further studies are required to formally elucidate these critical cellular processes, which are beyond the scope of our current study.

Finally, our studies reveal that mature SLE CD138+ ASC (Pops 3/5), exhibit a transcriptional program favoring cell survival, characterized by increased expression of anti-apoptotic genes and concurrent downregulation of pro-apoptotic mediators (Fig. 8). SLE ASC also display heightened levels of adhesion and homing receptors (PECAM1, ICAM1, CD69 and CXCR4), which facilitate migration and retention in protective survival niches. Specifically, PECAM1 and CD69 are likely to play crucial roles in the formation and migration of PC to the BM[83–86], together with CXCR4 promoting the latter function[87]. ICAM1, preferentially expressed in mature BM PC, is essential for homotypic aggregation and survival[88]. Notably, SLE ASC show elevated expression of Nur77 (NR4A1), Lyn, and Jun, signaling molecules induced by BCR activation, which have been demonstrated in murine genetic models to decrease PC survival and BM accumulation of PC[41,43,89]. Therefore, the overexpression of these molecules in SLE ASC may potentially represent an unsuccessful attempt to limit PC survival by antagonizing anti-apoptotic factors and counteracting cytokine responsiveness. It should be noted however that chronic Lyn stimulation induces autoantibody-mediated disease, a function suggesting a pathogenic role for the overexpression of this signaling kinase in human SLE ASC[42]. Overall, this profile consistently reflects the generation and maturation of SLE ASC within a highly inflammatory environment, distinguishing it from conventional protein vaccines. This disparity may underlie many of the observed differences between SLE and vax-HC in our study. The inflammatory milieu likely amplifies the epigenetic abnormalities inherent in SLE B cells, thus providing a potent stimulus for ASC development.

In conclusion, this research offers a comprehensive and original understanding of the phenotypic, molecular, and functional regulatory mechanisms underlying the generation, survival, and behavior of SLE ASC. These findings will significantly contribute to the informed selection and application of available therapeutic agents capable of targeting specific ASC populations based on their expression of surface markers, such as CD19 and CD138, as well as their chemokine receptors, survival factors, and regulatory TF. Particularly noteworthy is the potential of utilizing bifunctional agents, including monoclonal antibodies, CAR-T cells or T cell engagers, to selectively target ASC populations responsible for active autoantibody production. By leveraging these insights, future interventions can be designed to effectively modulate and manage SLE pathogenesis.

## Methods
### Human participants
All research was approved by the Emory University Institutional Review Board (Emory IRB numbers IRB00058515 and IRB00057983) and was performed in accordance with all relevant guidelines and regulations. Written informed consent was obtained from all participants.

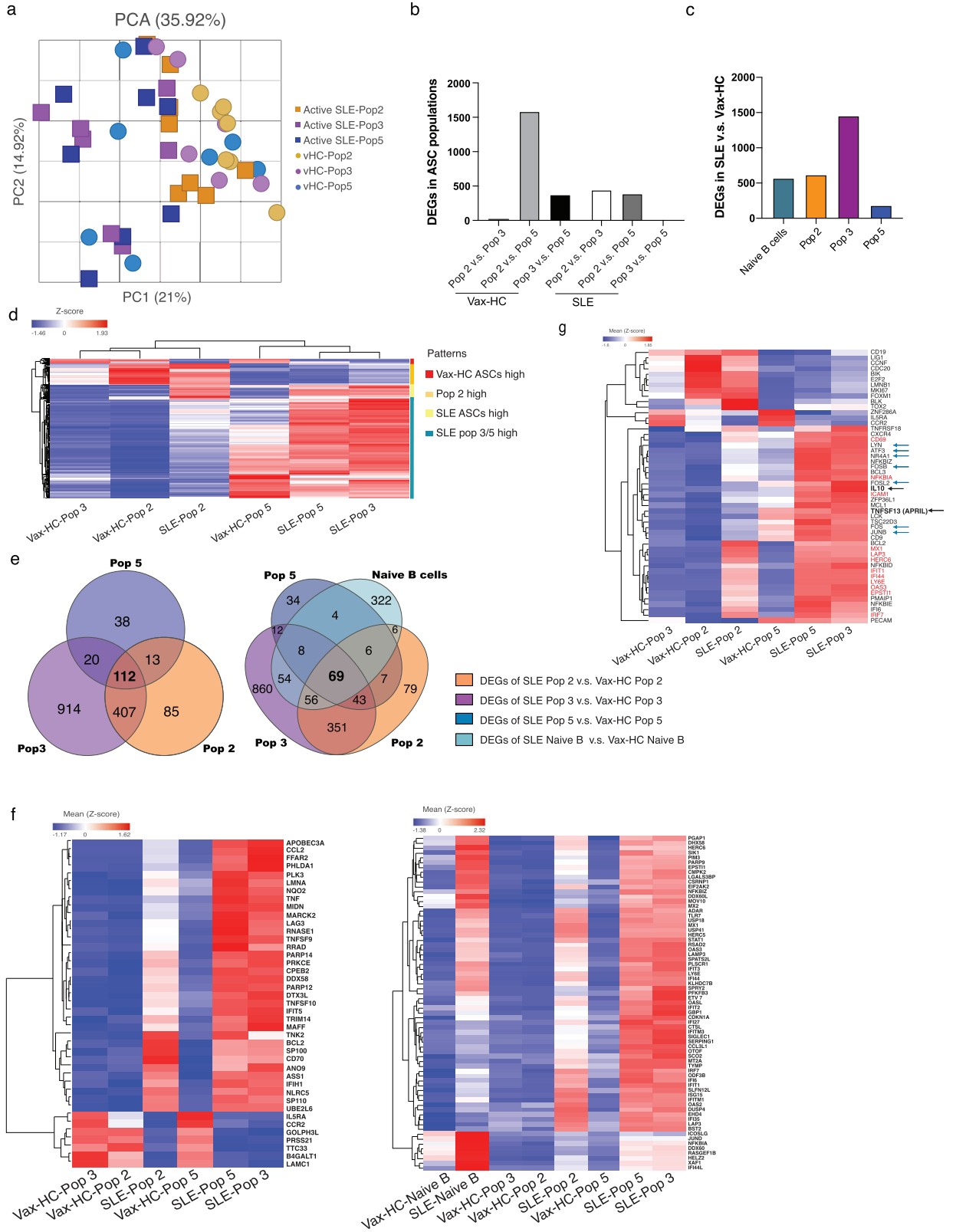

Healthy donors ($n = 45$) were recruited using promotional materials approved by the Emory University Institutional Review Board. Healthy subjects received the non-adjuvanted influenza vaccines ($n = 24$) as part of routine medical care. Subjects with SLE ($n = 176$) were recruited from Emory University Hospital and Grady Hospital in Atlanta, GA, USA. Peripheral blood mononuclear cells were isolated on

days 6–7 after vaccination for all vaccinated subjects. SLE patients fulfilled four or more criteria of the modified American College of Rheumatology classification (http://www.rheumatology.org/Practice/Clinical/Indexes/ Systemic_Lupus_Erythematosus_Disease_Activity_Index_SELENA_Modification/) and were routinely evaluated by expert rheumatologists at the Emory University Hospital and Grady Hospital.

**Fig. 6 | SLE ASC express a distinct transcriptome.** Peripheral ASC populations from 9 active SLE patients and 7 healthy subjects post influenza vaccination were sorted and their transcriptomes analyzed by RNA sequencing. **a** Principal component analysis (PCA) of differentially expressed genes (DEGs) (at least two-fold change of expression and FDR < 0.05) detected from all pairwise comparisons of the six groups. **b** DEGs between ASC pops within SLE and Vax-HC. Bar plots show DEGs within populations from same disease groups, including both active SLE patients and heathy subjects post vaccination, respectively. **c** Bar plots show DEGs between SLE and post-vax HC for each ASC population and naive B cells. **d** Heatmap of z-score normalized RPKM expression for all 2200 DEGs identified, and their distribution across four clusters indicated by the color code of the Y axis to the right: High expression in Vax-HC ASC; high expression restricted to pop2 whether SLE or HC; high expression in all SLE ASC; and high expression restricted to SLE pops 3 and 5. **e** Venn diagrams show DEG sharing between all ASC Pops identified from the comparison of SLE and vaccinated HC (left), and overlap of DEGs among all ASC Pops and naive B cells by comparing SLE and vaccinated HC (right). **f** Heatmap of z-score normalized RPKM expression for 43 overlapping DEGs shared between all ASC populations but not by naive B cells (left heatmap), and for the 69 genes shared between ASC populations and naive B (right heatmap). **g** Heatmap of z-score normalized RPKM expression for top DEGs of immunological relevance. IFN-stimulated genes are colored in red. Overexpression of IL-10 and APRIL is indicated in bold and black arrows. Transcription factors (TF) known to determine a large fraction of the SLE naive B cell transcriptome through epigenetic regulation are indicated by blue arrows.

Moderate-to-severe flare were classified according to the flare index of Safety of Estrogen in Lupus: National Assessment-Systemic Lupus Erythematosus Disease Activity Index (SELENA-SLEDAI).

## Multi-color flow cytometry and sorting

Cells were isolated from peripheral blood using Ficoll density gradient centrifugation and perioheral blood mononuclear cells (PBMC) were stained with the following anti-human antibody staining reagents: CD3-BV711 (clone: HIT3a, BD Biosciences, catalog number: 740768, dilution at 1:20); CD14-BV711 (clone: M5E2, BD Biosciences, catalog number: 740773, dilution at 1:20); IgD-FITC (IA6-2, BD Biosciences, catalog number: 555778, dilution at 1:5); CD19-PE-Cy7(clone: SJ25C1, BD Biosciences, catalog number: 341093, dilution at 1:5); CD27-APC-eFluor780 (clone: O323, Invitrogen, catalog number: 50-161-60, dilution at 1:20); CD38-V450 (clone: HIT2, BD Biosciences, catalog number: 561378, dilution at 1:20); CD138-APC (clone: 44F9, Miltenyi Biotec, catalog number: 130-127-977, dilution at 1:20); CXCR4-PE (clone: 12G5, BioLegend, catalog number: 306506, dilution at 1:20); CXCR3-PE (clone: G025H7, BioLegend, catalog number: 352706, dilution at 1:20); Blimp-1-PE (clone: 6D3, BD Biosciences, catalog number: 564702, dilution at 1:10); BCMA-PE (clone: 19F2, BioLegend, catalog number: 357504, dilution at 1:10), IL-6R-PE (clone: M5, BD Biosciences, catalog number: 561696, dilution at 1:20), BCL-2-PE (clone: Bcl-2/100, BD Biosciences, catalog number: 340576, dilution at 1:10). For cell sorting, cells were first enriched by Custom Stem Kit for negative slelection for CD66b, Glycophorin A, CD3 and CD14, followed by cell staining and sorting. FlowJo 10.7.1 was used for flow cytometry data collection and analysis. For cell sorting, approximately $1 \times 10^3$ to $5 \times 10^3$ were collected for each cell population by using FACSAria II (BD Biosciences).

## AIRR-seq

Sorted ASC populations (Pop 2, Pop 3, Pop 4 and Pop 5) from active SLE patients were used to extract the total cellular RNA by using the RNeasy Mini Kit (Qiagen, Inc. Valencia, CA) according to the manufacturer's protocol. Approximately 400 pg of RNA was subjected to reverse transcription using the iScript RT kit (BioRad, Inc., Hercules, CA). Resulting cDNA products were combined with 50 nM VH1- VH6 specific primers and 250 nM Cα, Cμ, and Cγ specific primers for isotype identification in a 25 μl PCR reaction using High Fidelity Platinum PCR Supermix (Life Technologies, Carlsbad, CA). Nextera indices were added, and products were sequenced on an Illumina MiSeq with a depth of approximately 50,000 sequences per sample. Sequences were quality filtered and aligned with IMGT.org/HighV-quest. Sequences were then analyzed for V region mutations and clonality using programs developed in-house and made previously available for public use. All clonal assignments were based on matching V and J regions, matching CDR3 length, and 85% CDR3 homology. Custom scripts and in-house developed analysis software were used with R, Matlab or Circos tools for visualization[8].

## In vitro plasma cell cultures

In vitro PC cultures were performed as follows: sorted ASC populations from active SLE patients and post-vaccination healthy donors were cultured in the secretome from BM-derived mesenchymal stromal cells in 96-well U-shaped bottom cell culture plates in 37 °C in a humid, 5% $CO_2$, 95% air (20% $O_2$) incubator for designated days. Cells were harvest for IgG Elispot to assess ASC survival. Exogenous factors including APRIL blocking antibody (human APRIL Antibody, Bio-techne), IL-10 blocking antibody (human IL-10 Antibody, Bio-techne), and their corresponding isotype controls (Monoclonal Mouse $IgG_1$ and Monoclonal Mouse $IgG_{2B}$, Bio-techne) were added to cultures at day 0 post cell sorting, and their concentrations were titrated and optimized based on manufacture's recommendations. Following the incubation, supernatant was collected for ELISA and cells were harvested for ELISPOT assays run in triplicate.

## Total IgG ELISPOT assay

Sorted ASC populations were added to 96-well ELISPOT plates (MAIPS4510 96 well) pre-coated with anti-human IgG (5 μg/ml, Jackson Immunoresearch). After overnight incubation, wells were washed, and bound antibodies were detected with alkaline phosphatase-conjugated anti-human IgG antibody (1μg/ml) and developed with VECTOR Blue Alkaline Phosphatase Substrate Kit III (Vector Laboratories). Spots in each well were counted using the CTL immunospot reader (Cellular Technologies Ltd). Results were expressed as the ratio of antigen-specific spots/total IgG spots.

## Enzyme-linked immunosorbent assay (ELISA)

Costar assay high-binding plates were coated with 4 μg/ml ELISA capture antibody IL-10 (Human IL-10 Antibody, Bio-techne) or APRIL (Human APRIL Antibody, Bio-techne) overnight, and then washed with PBS with 0.1% Tween-20 and blocked with SuperBlock blocking buffer for 45 min. Plates were next incubated with culture supernatants for 60 min. Bound antibodies were then detected with paired detection secondary antibodies at 1.5 μg/ml followed by chromogenic detection with KPL BluePhos Microwell Phosphatase Substrate Kit.

## RNA sequencing and analysis

RNA sequencing was performed as follows, populations including Pop 2 (CD19+ CD138−), Pop 3 (CD19+ CD138+), Pop 5 (CD19− CD138+), and naive B cells (CD19+ IgD+ CD27−), isolated from active SLE patients or influenza-vaccinated healthy subjects were sorted directly into lysis buffer. RNA was isolated using the AllPrep DNA/RNA Mini kit (Qiagen) according to the manufacturer's instruction. Total RNA (50 pg) was used as input for the ten cycles of PCR amplification by using SMART-seq v3 cDNA synthesis kit (Takara). Libraries were quantified by qPCR and size distribution was examined using a Bioanalyzer before pooling and sequencing on Illumina HiSeq 2500 sequencer using 50-bp paired-end sequencing.

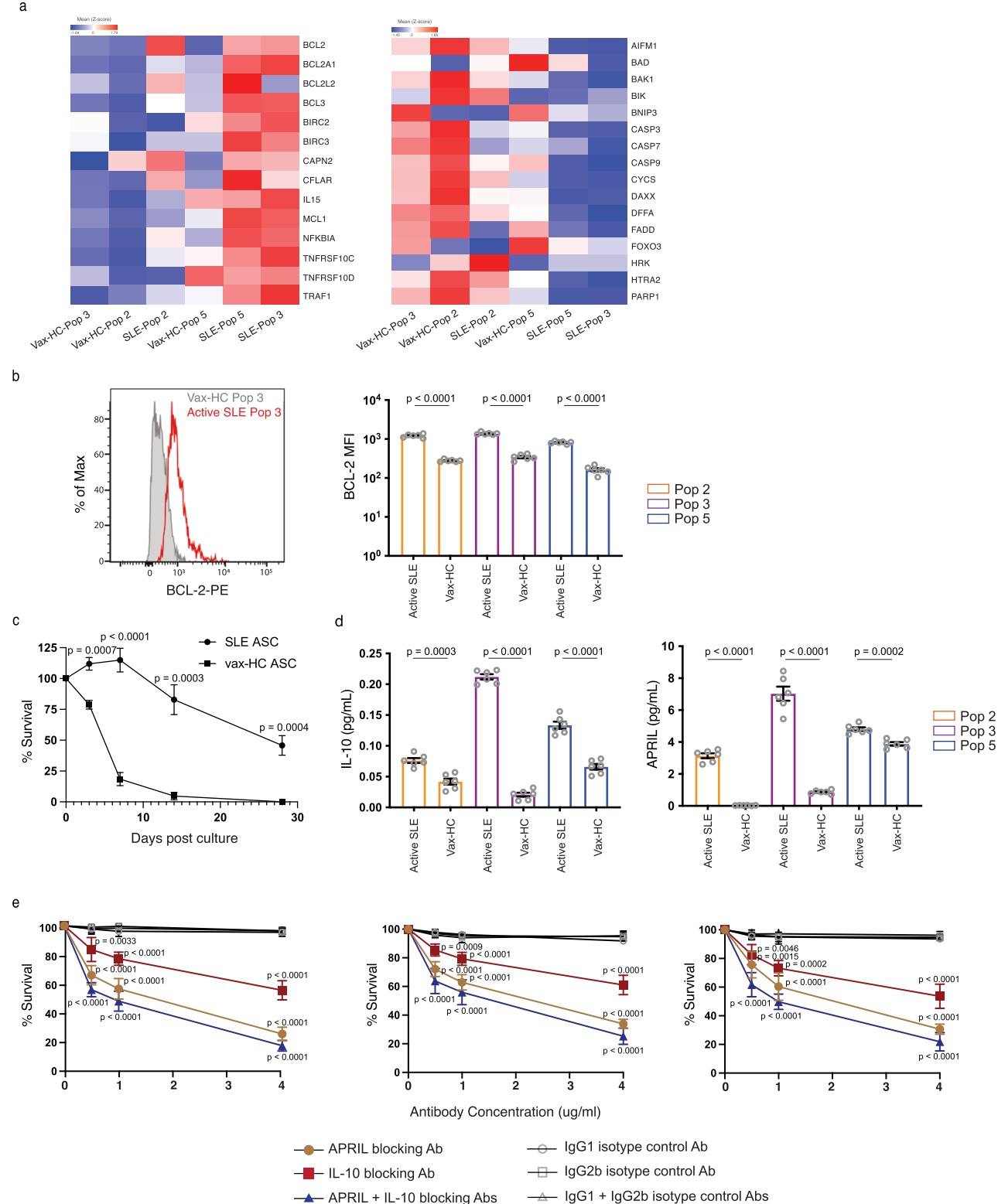

## RNA sequencing data analysis

Raw fastq files were mapped to the human genome using TopHat2 v.2.0.13 with the hg19 version and the UCSC KnownGene reference transcriptome. All duplicate reads were excluded with PICARD v.1.127, followed by quantification with Partek Flow Genomics Suite (Partek Inc.) using the union model in HTSeq with the default setting, and library-size normalization using the DESeq2 with default parameters. Differential gene expression analysis was determined with a generalized linear effect model. For disease differences covariates included cell type and patients; for cell type differences covariates included disease status and patients. Genes with at least a two-fold change and an FDR value ≤ 0.05 among comparison were considered significant and termed as DEGs. Quantile normalized RPKM values were used in the heat maps and z-score values were plotted for genes visualization.

## Cytospin of sorted ASC populations

Cytospin was performed as follows: FACS-sorted ASC populations from PBMC were pelleted at 1300 rpm for 5 min on the Cytospin 4

**Fig. 7 | SLE ASC Express a pro-survival transcriptome and enhanced intrinsic and extrinsic in vitro survival. a** Heatmap of z-score normalized RPKM expression for anti-apoptotic genes (left) and pro-apoptotic genes (right) in ASC Pops from SLE and healthy subject post vaccination. Data represent the mean expression for each group. **b** BCL-2 protein expression on ASC populations from SLE ($n = 6$) and vaccinated HC ($n = 6$) by intracellular flow cytometry is shown in a representative example (left) and compiled data (right). Data are shown as mean ± SEM. Statistical significance was assessed using Student's $t$ test among disease groups within the same ASC population. **c** Survival differences determined by SLE ASC intrinsic properties were ascertained using cultures of sorted CD19⁺ ASC from SLE ($n = 5$) and post-vax HC ($n = 5$) in MSC media in the absence of exogenous APRIL, IL-10 or other cytokines. In vitro ASC survival was measured on day 0, 3, 7, 14 and 28 post culture by IgG ELISpot and normalized to the IgG ELISpot numbers of sorted ASC prior to culture. Data are shown as mean ± SEM. Statistical significance was assessed using Student's $t$ test between SLE patients and vaccinated healthy subjects. **d** Sorted ASC Pops 2, 3, and 5 from active SLE ($n = 6$) and post-vax HC ($n = 6$) were cultured in MSC media, and the supernatant harvested on day 3 to detect the secretion of IL-10 and APRIL by ELISA and normalized to the ELISpot numbers of each population on day 3. Data are shown as mean ± SEM. Statistical significance was performed using Student's $t$ test within the same ASC population. **e** Sorted ASC Pops 2, 3, 5 from active SLE ($n = 5$) were cultured in MSC media with neutralizing antibody against IL-10, APRIL, or both, or their corresponding isotype controls at concentrations of 0.5 ug/ml, 1 ug/ml and 4 ug/ml, and cell survival was measured by IgG ELISpot on day 3, normalized to ASC cultured in the absence of antibodies, and compared with corresponding isotype controls. Data are shown as mean ± SEM. Statistical significance was assessed using Student's $t$ test between neutralizing antibodies and their corresponding isotype controls at the same concentration. $p$ values are shown on plots. Source data are provided as a Source Data file.

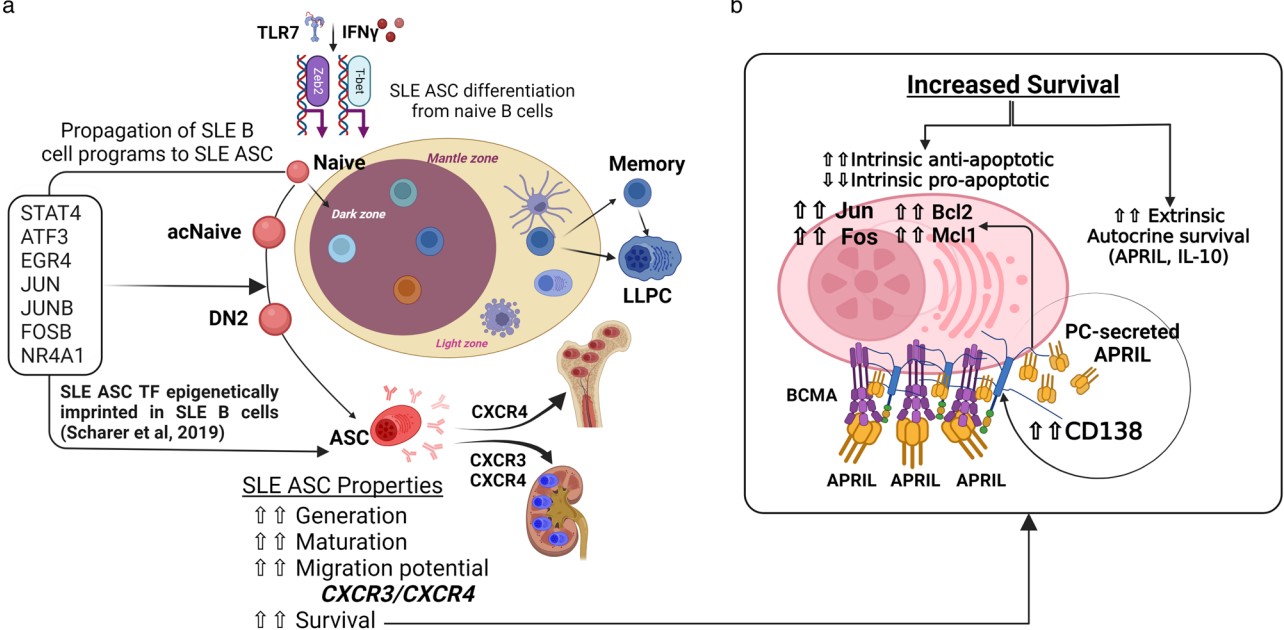

**Fig. 8 | The sustained survival of circulating SLE ASC results from a combination of intrinsic transcriptional programs and extrinsic factors. a** The coordinated overexpression of CXCR3 and CXCR4 renders circulating SLE ASC highly prone to migration and retention within anatomical locales characterized by a prominent inflammatory milieu in SLE. **b** Schematic representation of the intrinsic and extrinsic mechanisms of prolonged survival of SLE ASC. Intrinsic mechanisms include the transcriptional regulation of apoptotic programs and extrinsic mechanisms include autocrine APRIL and IL-10 forward loops, with the effects of APRIL further enhanced by the overexpression of CD138, which facilitates its interaction with BCMA, an essential receptor for PC survival.

(Thermo Scientific, Waltham, MA). Approximately 5000 cells per population were dried overnight on albumin-coated slides and stained with Wright-Giemsa stain and analyzed using Image J 1.53.

### Transmission electron microscopy

FACS-sorted ASC populations were pelleted by centrifugation at 500 × $g$ for 5 min. Pellets were then resuspended in PBS with $1 \times 10^6$ erythrocytes to help visualize the pellets during the TEM processing. The pellets were then fixed (overnight, at 4 °C) using 2.5 M glutaraldehyde and then placed in 0.1% osmium tetroxide in 0.1 M phosphate buffer (pH 7.4) for 1 h. Dehydration of pellets was performed by sequential incubation in ethanol solutions at different concentrations (25%, 50%, 75%, 95% and 100% EtOH). The pellets were then infiltrated, embedded, and polymerized using Eponate 12 resin (Ted Pella Inc.). Leica Ultracut S ultramicrotome was used to cut sections ~70 nm thick, which were then stained with 5% uranyl acetate and 2% lead citrate. JEOL JEM-1400 TEM (JEOL Ltd) with Gatan US1000 2k × 2k CCD camera (Gatan) were used for imaging.

### Statistical analysis

Graphpad Prism version 8.2.0 software was used for statistical analysis. Statistical comparisons between two groups were performed using Student's $t$ test. Statistical comparisons between more than two groups were determined using Kruskal–Wallis test followed by Dunn's test for multiple pairwise comparisons.

### Reporting summary

Further information on research design is available in the Nature Portfolio Reporting Summary linked to this article.

## Data availability

The AIRR sequencing and RNA sequencing datasets for ASC populations were deposited at Genome Expression Omnibus under the accession number GSE235660. Source data are provided with this paper.

## Code availability

Custom scripts and in-house developed analysis software were used with R, Matlab or Circos tools for visualization AIRR sequencing data,

and the scripts are available on GitHub at https://github.com/chenwr56/airr.

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

## Acknowledgements

The authors are indebted to our patients, healthy volunteer, and our team of clinical coordinators who together made this study possible. We would like to thank members of the Sanz and Lee laboratories for their insights and contributions into this study. This study was supported by the Emory University School of Medicine Flow Cytometry Core, the Emory Pediatric Flow Cytometry Core, the Emory Integrated Cellular Imaging core, the University of Alabama at Birmingham High Resolution Imaging Shared Facility and Distinguished Innovator Award from the Lupus Research Alliance. This research project was supported by National Institutes of Health grant: 2P01-AI125180-06.

## Author contributions

Conceptualization: W. Chen, S. Hong, I. Sanz, and F.E-H.Lee; Methodology: W. Chen, S. Hong, F. A. Anam, S. Hicks; Software: W. Chen, C. Tipton, J. Hom, M. Woodruff; Validation: S. Jenks; Formal Analysis: W. Chen, C. Tipton, C. Scharer, T. Mi; Investigation: W. Chen, S. Hong, C. Tipton; Resources: S. Jenks, K. S. Cashman, X. Wang, C. Wei, A. Khosroshahi, S. Lee, S. Kyu, D. C. Nguyen, L. Harton, Y. Wang, R. Bugrovsky, Y. Ishii, C.E.Faliti; Writing Original Draft: I. Sanz, W. Chen, S. Jenks; Visualization: W. Chen; Supervision: I. Sanz, and F.E-H.Lee; Funding Acquisition: I. Sanz, and F.E-H.Lee; Writing Review and Editing: all authors reviewed, edited, and approved the manuscript.

## Competing interests

The authors declare no competing interests.
