## [Peer Review File · Nature Communications]

Distinct transcriptomes and autocrine cytokines underpin maturation and survival of antibody-secreting cells in systemic lupus erythematosusEditorial Note: This manuscript has been previously reviewed at another journal that is not operating a transparent peer review scheme. This document only contains reviewer comments and rebuttal letters for versions considered at *Nature Communications* .

REVIEWERS' COMMENTS

Reviewer #1 (expert in B cells and plasma cells in inflammatory diseases):

No further main concerns.

Minor points

Line 536: please correct "adaptive and adaptive immune....."

[Reviewer #1 was also asked to comment on the responses to Reviewer #3, who was no longer available for review]

please find below my opinion on the authors response on the concerns of reviewer 3.

The main concern of reviewer 3 was that the manuscript failed to show qualitative differences between plasma cells in active SLE and controls. Reviewer 3 had some specific points in this regard, which were adequately addressed in the response letter, but only partly in the revised manuscript.

Particularly, reviewer 3 had concerns about the quality of the BCMA and BLIMP1 staining. The authors argue that in their hands, BCMA and BLIMP1 staining are rather distinctive (# 35 of the point by point response). To proof their argument they show the original data in the rebuttal letter (Figure R3). The staining is convincing. To ensure that the response to reviewer 3 is also adequately addressed in the paper, these data should be also included in the revised version of the manuscript.

Likewise, a graphical summary describing how hematopoietic cells are mechanistically unique in active SLE should be included in the revised manuscript (# 36 of the point by point response), which could be based on Figure R5 shown in the point by point response.

Other points raised by reviewer 3 were already adequately addressed in the revised version of the manuscript. These include minor issues such as the quality of the figures etc., which have now been improved in the revised version.

Reviewer 3 also wondered if CD19 negative plasma cells in SLE resemble mature emigrants from the bone marrow, rather than to be a SLE specific population. In their response, the authors provided several lines of evidence from published papers that CD19 negative plasma cells in SLE are not bone marrow emigrants (and cited a very recent paper relevant in this context). Even more convincingly, the authors argued that the CD19 negative plasma cells in their study expressed the proliferation marker Ki67+, excluding the possibility that these cells are mature (non-proliferating) bone marrow emigrants. The response seems to be convincing to me.

In addition, reviewer 3 asked about original flow data of CXCR3/CXCR4 expression. However, these data were already shown in the supplemental figure 2 of the original manuscript (see Point by point response # 34).

In response to one of the concerns, data on Bcl-2 protein expression are now included in the revised version of Fig 7, which is relevant for plasma cell survival (# 37). This is important new data, since the increased and distinct expression of anti-apoptotic Bcl-2 in SLE plasma cells strongly supports the conclusion made in the manuscript that SLE plasma cells are qualitatively different from plasma cells of healthy donors.

Therefore I believe that all concerns of reviewer 3 are adequately addresses. But I guess that reviewer 3 would like to have included the data shown in Figure R3 of the rebuttal letter and a respective graphical summary as described above, in the manuscript.

Reviewer #2 (expert in humoral immunity):

The manuscript is significantly improved. I am very impressed by the authors extensive and thoughtful response to all questions and comments. I think this work makes an important contribution to the characterisation of SLE and includes very sophisticated experimental and analytical approaches that will be benchmarks for the field.